# Identification and Study of the Action Mechanism of Small Compound That Inhibits Replication of Respiratory Syncytial Virus

**DOI:** 10.3390/ijms241612933

**Published:** 2023-08-18

**Authors:** Anna A. Shtro, Artem M. Klabukov, Anzhelika V. Garshinina, Anastasia V. Galochkina, Yulia V. Nikolaeva, Tatyana M. Khomenko, Danila E. Bobkov, Aleksey A. Lozhkov, Konstantin V. Sivak, Kirill S. Yakovlev, Andrey B. Komissarov, Sophia S. Borisevich, Konstantin P. Volcho, Nariman F. Salakhutdinov

**Affiliations:** 1Smorodintsev Research Institute of Influenza, Professora Popova str, 15/17, 197376 St. Petersburg, Russia; temaklab@gmail.com (A.M.K.); garshininaang@gmail.com (A.V.G.); nastyagalochkina@gmail.com (A.V.G.); ulechka.s89@gmail.com (Y.V.N.); bobkovde@yandex.ru (D.E.B.); aswert6@mail.ru (A.A.L.); kvsivak@gmail.com (K.V.S.); kirikus-fly@yandex.ru (K.S.Y.); abkomissarov@gmail.com (A.B.K.); 2Vorozhtsov Novosibirsk Institute of Organic Chemistry, Acad. Lavrentyev av. 9, 630090 Novosibirsk, Russia; chomenko@nioch.nsc.ru (T.M.K.); volcho@nioch.nsc.ru (K.P.V.); anvar@nioch.nsc.ru (N.F.S.); 3Institute of Cytology Russian Academy of Science, Tikhoretsky av., 4, 194064 St. Petersburg, Russia; 4Ufa Chemistry Institute of the Ufa Federal Research Center, 71 Octyabrya pr., 450054 Ufa, Russia; monrel@mail.ru

**Keywords:** coumarin, antiviral activity, cytotoxicity, respiratory syncytial virus

## Abstract

Respiratory syncytial virus (RSV) is known to cause annual epidemics of respiratory infections; however, the lack of specific treatment options for this disease poses a challenge. In light of this, there has been a concerted effort to identify small molecules that can effectively combat RSV. This article focuses on the mechanism of action of compound K142, which was identified as a primary screening leader in the earlier stages of the project. The research conducted demonstrates that K142 significantly reduces the intensity of virus penetration into the cells, as well as the formation of syncytia from infected cells. These findings show that the compound’s interaction with the surface proteins of RSV is a key factor in its antiviral activity. Furthermore, pharmacological modeling supports that K142 effectively interacts with the F-protein. However, in vivo studies have shown only weak antiviral activity against RSV infection, with a slight decrease in viral load observed in lung tissues. As a result, there is a need to enhance the bioavailability or antiviral properties of this compound. Based on these findings, we hypothesize that further modifications of the compound under study could potentially increase its antiviral activity.

## 1. Introduction

Respiratory syncytial virus (RSV) is a highly contagious virus belonging to the Pneumoviridae family, specifically the genus Orthopneumovirus. It is a significant contributor to illness and discomfort in newborns, young children, and adults. This virus is responsible for causing acute respiratory tract infections (ALRIs), which can potentially be fatal, especially for infants.

Among young children, RSV is the most common viral pathogen associated with ALRIs. Although its role in ALRI mortality is not exactly clear, in 2005 alone, RSV was responsible for the deaths of 66,000 to 199,000 children under the age of five, with a staggering 99% of these deaths occurring in developing countries [1].

RSV is highly prevalent, with approximately 50% of infants becoming infected within their first year of life, and nearly all individuals having some form of contact with the virus by the age of two [2]. Certain groups, such as children with chronic illnesses and individuals with weakened immune systems (including the elderly and those who have recently undergone organ transplantation), are particularly vulnerable to severe complications.

Currently, the only available strategy of protection against RSV is passive immunization, which involves the administration of an RSV-specific monoclonal antibody called palivizumab. Additionally, two recently FDA-approved vaccines, ABRYSVO by Pfizer and Arexvy by GlaxoSmithKline Biologicals, offer passive immunization for individuals aged 60 and above only [3,4]. To further combat this infection and treat RSV-related diseases, potential future strategies could include vaccinating newborns and children, as well as passive immunization with a vaccine or antibodies that have an extended half-life. It is of utmost importance to develop new strategies to effectively combat this widespread infection and alleviate the burden it places on individuals and communities.

In 2021, primary screening was conducted to assess the antiviral activity of monoterpene-coumarin conjugates against RSV [5]. This study led to the identification of a new structural type of potent inhibitors of RSV replication. The leader compound **19c**, now having a code name K142 (Figure 1) ((3-(((1S,5R)-6,6-dimethylbicyclo [3.1.1]hept-2-en-2-yl)methoxy)-7,8,9,10-tetrahydro-6H-benzo[c]chromen-6-one)), demonstrated good activity against both RSV A and B, with a selective index of more than 70 and an IC_50_ ranging from 4.9–5.1 μM [5].

Recently, further research [6] revealed the antiviral activity of structural analogs of K142, which contained 4-aryl-substituted coumarin cores conjugated with monoterpene fragments. Although the most active compounds were found to be less potent than K142, they exhibited slightly better selectivity indexes [6]. Therefore, coumarin-monoterpene conjugates hold promise for the development of new anti-RSV agents.

Based on time-of-addition experiments and molecular modeling, the RSV F protein was suggested as a potential target for K142 [5]. The current work aims to delve deeper into the mode of action of K142 and test its efficacy in an in vivo experiment.

## 2. Results

### 2.1. Interaction of RSV with the Host Cell at Various Stages of the Life Cycle in the Presence/Absence of K142 Using Confocal Microscopy

Our previous study demonstrated that K142 most likely targets the early stages of the virus life cycle [5]. To gather further data, we employed confocal microscopy to visualize the various stages of the viral life cycle in the presence of K142. 

In order to conduct the experiment, the compound was initially dissolved in DMSO and then adjusted to a final concentration of 100 µg/mL. As for the control wells, instead of administering the compound, a culture medium containing a similar concentration of DMSO was used. 

Following this, the cell culture was fixed after two specific time intervals: firstly, after a 2-h period of virus penetration into the cell, and secondly, after a 3-day period of syncytium formation (formation of cell clusters). 

Upon completion of the necessary incubation period, the cells were washed with PBS and subsequently fixed with a 10% formalin solution in PBS. They were then stained with mouse primary antibodies against RSV and incubated with rabbit secondary anti-mouse antibodies that were conjugated with a fluorescent label. The resulting preparations were visualized using a confocal microscope, as depicted in Figure 2 and Figure 3. 

From Figure 2, it is evident that cells in the control group are covered with viral particles, which completely envelop the cell membrane and also form fainter filamentous structures extending into the cell. Conversely, in the presence of the compound K142, fragmented viral antigen on the membrane can be observed, without continuous coverage of the cell membrane. Furthermore, the overall amount of antigen is significantly less than in the control group.

Additionally, we fixed the cells on the 3rd day after infection, wherein most of the cells had already undergone fusion. However, complete cell death and the formation of a comprehensive cytopathogenic effect are not observed at this stage (refer to Figure 3).

The table of images depicts the formation of a multidimensional structure consisting of multinuclear syncytia in the control group. These syncytia are not only present within the plane of preparation but also extend above it. Detailed analysis reveals that the entire surface of the syncytium is densely covered with viral particles.

The usage of compound K142 reduces the number of syncytia and viral antigen signal. Statistical analysis of the results demonstrated a significantly lower intensity in the cell culture exposed to the test compound at both time points (refer to Figure 4).

From these findings, it can be seen that K142 has an impact on the process of virus penetration into the cell, as well as the formation of fully developed syncytia within the infected cell monolayer. This aligns with previous studies [5] regarding the compound’s mechanism of action, which suggest its effect on the viral proteins F and G is responsible for these processes.

### 2.2. Evaluation of the Antiviral Efficacy of K142 In Vivo

In the subsequent phase of our research, we evaluated the effectiveness of K142 against RSV in Balb/c mice. Considering the pharmacokinetic data of K142 [7], we opted for oral administration at a dosage of 50 mg/kg. Additionally, we explored the drug’s activity through an alternative mode of administration intranasally.

Due to the lack of specific pharmacokinetic data for this route, we employed two dosages that differ by a factor of 10. Since topical application leads to significantly higher concentrations of the test substance in the tissues where the virus replicates directly, we reduced the maximum dosage of K142 to 10 mg/kg. The second dosage was 1 mg/kg. As a reference drug, ribavirin was administered at a dosage of 50 mg/kg.

The study revealed that infection did not result in a significant weight loss in the infected animals across all groups, except for one. The changes in body weight of the animals over time are illustrated in Figure 5. 

Based on the data presented in the figure, infected animals generally did not experience weight loss, except for mice that received the intranasal administration of K142 at a dose of 10 mg/kg. This weight loss may be attributed to difficulties in nasal breathing post-drug administration, as the drug was in the form of a viscous suspension, potentially impeding normal airflow through the nasal passages. It is worth noting that mice in this particular group exhibited increased excitability and aggressiveness, which could be indicative of stress resulting from temporary breathing difficulties. No such effects were observed in the other groups.

On the 6th day after infection, mice were euthanized via cervical dislocation, and their lungs were analyzed. For each sample, the absolute quantity of viral antigen in the sample and its concentration per gram of lung tissue were determined. Subsequently, the statistical significance of the differences between the groups was analyzed. The obtained data are presented in Table 1 and Figure 6.

From the data presented in Table 1 and Figure 6, it can be observed that the viral load in the lungs of mice treated with the reference drug ribavirin showed a statistically significant decrease compared to the control group. This indicates the suitability of the model employed and its relevance for evaluating the antiviral activity of experimental drugs against RSV.

While the use of K142 in all three groups did not result in a statistically significant decrease in viral load, there was a noticeable trend toward reduction in viral load across all groups receiving the agent. The most significant decrease was observed in the group that received intranasal administration of K142 at a dose of 1 mg/kg.

Histological examination of the lung tissue (Figure 7) revealed the presence of bronchogenic inflammatory foci in all groups under study. These foci were primarily located around the bronchi, with limited extension into the underlying lung tissue. Additionally, the examination identified erythrocytes present in the alveolar lumens in several samples.

Furthermore, all samples exhibited a consistent pattern of lung tissue damage, characterized by varying degrees of bronchiolitis and bronchogenic interstitial pneumonia. The walls of the bronchi and bronchioles showed swelling, along with significant infiltration of lymphocytes in the peribronchial area. Necrosis and desquamation of the ciliated epithelium into the bronchial lumen were also observed. The interalveolar septa appeared thickened and were abundantly infiltrated with lymphocytes.

The most severe and widespread lung lesions were observed in the placebo group and the group receiving oral administration of K142. Conversely, the groups treated with intranasal administration of K142 at a dose of 10 mg/kg and the group receiving the reference drug ribavirin exhibited a more favorable course of disease.

In summary, considering the various indicators employed in this study, such as the relative and absolute lung weights, viral load in lung tissue, and severity of the infectious process, it can be concluded that the K142 drug exhibits a weak antiviral activity against the respiratory-syncytial virus, with the most prominent effect observed in the groups treated with the drug via intranasal administration.

### 2.3. Simulation Results

The primary aim of conducting molecular dynamics simulations was to evaluate the behavior of the K142 ligand [5] within the binding site. The model system was designed in such a way that the transmembrane domain of the protein was immersed in the membrane (Figure 8). It is believed that in molecular dynamics simulations involving such proteins, the presence of a membrane is crucial for accurately positioning the secondary protein structure throughout the simulation. In this case, we are referring to the heptad repeats of the F-protein, which consists of α-helices adjacent to the viral membrane (see Figure 8A).

Based on the analysis of the RMSD perturbation graph, it can be observed that the ligand–protein system reached equilibrium by the end of the molecular dynamics simulations (Figure 8B). Throughout the entire simulation, the ligand remained within a symmetrical binding site and did not diffuse into the solvent. However, the geometric parameters of the ligand underwent notable changes, particularly when compared to the initial position (Figure 8A,B). Of particular interest is the tetrahydro-benzochromen-one fragment, which is highlighted by a sphere in Figure 8 and was initially located at the starting position corresponding to the beginning of the simulation.

While located within the binding site, the ligand primarily engages in hydrophobic interactions with a symmetrical arrangement of amino acid residues (a.a.r.) in the binding site. Specifically, the ligand interacts with three protomers via hydrophobic π-π cation stacking interactions with Phe137, Phe140 fusion loops, and Phe488 heptad repeat (HRB) (Figure 8). These interactions are maintained for 80% of the simulation time.

In addition, short-lived hydrogen bonds and water-mediated contacts were observed. The longest interaction was recorded with Arg339. Overall, over the course of the molecular dynamics simulations, the ligand forms at least eight intermolecular contacts with the a.a.r. of the binding site.

Based on the results obtained from the molecular dynamics simulations, several observations can be made. Firstly, the ligand has the ability to bind within the trimeric structure of the F-protein, specifically in a hydrophobic cavity located between the fusion peptide and the heptad α-repeat helix. This binding site is believed to play a crucial role in the inhibitory effect of the compound.

During the simulations, the ligand establishes a series of intermolecular contacts, primarily through π-π stacking interactions between the aromatic rings of the ligand and key amino acid residues, such as phenylalanine (Phe). These contacts are maintained throughout the entire simulation.

An interesting observation is the potential role of Phe488 in the inhibitory effect of the compound. Here we can assume that the inhibitory effect of the compound under study may be due to the effect of the side chains of Phe488 on the conformation, as described in [8].

## 3. Discussion

Moving on to the discussion of the antiviral activity and in vivo efficacy of the K142 compound, the mechanism was investigated. Confocal microscopy analysis showed that K142 significantly reduces the intensity of virus penetration into the host cell. Additionally, it demonstrated a reduction in the formation of syncytia, which are multinucleated cells formed from the fusion of infected cells. Furthermore, pharmacological modeling studies provided supporting evidence for the interaction between K142 and the RSV F-protein. 

Overall, the results obtained from both molecular dynamics simulations and experimental studies suggest that the K142 compound exhibits antiviral activity by interfering with the viral entry process and reducing syncytia formation. The binding of K142 to the F-protein plays a crucial role in these activities, specifically through its interactions with the hydrophobic cavity and key amino acids such as Phe488. Further investigations are needed to fully understand the molecular details of the inhibitory mechanism and to assess the potential of K142 as a therapeutic agent against RSV.

The in vivo study revealed a weak antiviral activity of K142 against RSV infection. However, there was a tendency towards a decrease in viral load in lung tissues. In conclusion, it can be stated that K142 exhibits poor efficacy when administered intranasally and requires improvement in terms of bioavailability and antiviral properties.

Currently, the number of etiotropic drugs targeting RSV is limited, and treatment for RSV infection in children is primarily supportive. Ribavirin is an exception, although it is reserved for severe cases due to its significant side effects.

The search for new, effective, and safe drugs against RSV is ongoing. According to the clinicaltrials.gov website, there are currently 374 clinical trials underway related to respiratory syncytial infection, including studies on drugs that target the early stages of the virus life cycle, similar to K142. Palivizumab, a humanized antibody against the RSV F protein, is already approved for the prevention of RSV infection, but it is not used for the treatment of established diseases. 

In this context, working with small molecules appears to be more promising. For example, the compound BMC-433771, developed by Bristol Myers Squibb selectively inhibits the F protein and has demonstrated activity in animal models [9], albeit in a prophylactic treatment regimen. Its derivative, Rilematovir (JNJ-53718678), is currently undergoing clinical trials [10]. In vitro experiments have shown high activity of Rilematovir, with an EC_50_ value of 9.3 μM [11]. Comparatively, the efficacy of K142 in vitro conditions shows similar indications. However, it is challenging to compare activity in animal models due to the variation in animal species used.

In addition, a screening process involving over 130,000 compounds led to the identification of the agent JNJ-2408068 by the pharmaceutical company Johnson & Johnson/Tibotec. This compound displayed high in vitro activity; however, experiments using an animal model yielded unsatisfactory results. Subsequent modifications of the JNJ-2408068 backbone led to the creation of TMC353121, which exhibited improved pharmacological properties. TMC353121 has shown effectiveness in reducing viral load and lung pathology, even when administered as early as 2 days after infection in mice. Successful experiments have also been conducted on primates [12]. 

Hence, it is evident that even major pharmaceutical companies encounter situations where a drug with promising in vitro antiviral activity does not perform satisfactorily in an animal model. However, through comprehensive study and further modifications, structurally analogous compounds can be developed to achieve higher efficiency. Research with substances similar in structure to K142 will continue until the discovery of compounds that demonstrate effectiveness in both in vivo and in vitro settings. This research contributes to the ongoing efforts in developing effective treatments against RSV infections.

## 4. Materials and Methods

### 4.1. Materials and Methods for All Experiments

#### 4.1.1. Compound

Compound under study, K142 ((3-(((1S,5R)-6,6-dimethylbicyclo [3.1.1]hept-2-en-2-yl)methoxy)-7,8,9,10-tetrahydro-6H -benzo[c]chromen-6-one)) was obtained from Vorozhtsov Novosibirsk Institute of Organic Chemistry, Siberian Branch of the Russian Academy of Sciences, Novosibirsk. Dosage form: substance. Storage conditions: at a temperature of +4 °C. 

#### 4.1.2. Virus

We utilized the A2 strain of the human RSV obtained from the laboratory of chemotherapy for viral infections at the Smorodintsev Influenza Research Institute. The strain was received on 4 January 2018 from the Laboratory for Biotechnology of Diagnostic Preparations. Subsequently, it was accumulated in a HEp-2 cell culture and stored in aliquots at a temperature of −80 °C.

#### 4.1.3. Cell Culture

We employed the HEp-2 cell culture, derived from HeLa cells, as it is known to be highly sensitive and permissive to the RSV. The culture was obtained from the working collection of the Laboratory of Chemotherapy for Viral Infections at the Smorodintsev Influenza Research Institute.

#### 4.1.4. Culture Medium

The culture medium consisted of 100 mL of DMEM medium (DMEM nutrient medium with a glucose concentration of 4.5 mg/mL (Biolot, St. Petersburg, Russia). Additionally, 1 mL of an antibiotic solution (penicillin-streptomycin), 2% of FBS serum, and 4 mM of L-glutamine were added to the medium. 

#### 4.1.5. Preliminary Assessment of Virus Titer in HEp-2 Cell Culture

The infectious activity of the virus was measured in a HEp-2 cell culture with 50–70% monolayer formation, 24 h after inoculation on the plates. A series of 10-fold dilutions (10^−1^–10^−7^) was prepared from a virus sample using DMEM medium with glutamine (Biolot, St. Petersburg, Russia) supplemented with 2% fetal bovine serum (Biolot, St. Petersburg, Russia) and 20 µg/mL ciprofloxacin (Sintez, Kurgan, Russia). These dilutions were then added to the wells of a 96-well plate with cells. The plates were incubated at 37 °C for 1 h in a 5% CO_2_ atmosphere. Afterward, the virus was washed off with 100 µL of supporting medium, and 100 µL of the same medium was added to each well. The plates were then incubated for 6 days until cytopathic effect (CPE) appeared in the control virus wells. Virus titers in the samples were determined by ELISA, as described below.

#### 4.1.6. Cell-ELISA

The enzyme immunoassay, specifically the cell-ELISA, was performed on the cell culture. To start with, the cell culture was fixed by exposing it to cold 80% acetone for 15 min at −20 °C. Following this, the culture was thoroughly washed using phosphate-buffered saline (Biolot, St. Petersburg, Russia) with the addition of Tween 20 (Orgsintez, Nizhniy Novgorod, Russia), ensuring a concentration of 0.05%. Subsequently, a solution containing primary mouse antibodies obtained as described in [13] targeting the RSV F protein was applied to the culture. The culture was then incubated for 2 h with continuous stirring at room temperature. After the incubation period, the cells were rinsed again with a buffer solution, and secondary anti-mouse antibodies were introduced. Another round of incubation for 2 h with continuous stirring followed this step. The remaining antibodies were washed off, and a substrate-chromogenic mixture containing tetramethylbenzidine was applied. The reaction was allowed to proceed for 5 min before halting it with 0.1 M sulfuric acid. To quantify the optical density of the resulting solution, measurements were taken at a wavelength of 450 nm. Wells with an absorbance value two or more times higher than that of the control cell wells were considered to be contaminated. Finally, the virus titer was determined using the Reed and Muench method.

### 4.2. Registration of the Viral Life Cycle in the Presence of K142 by Confocal Microscopy

Cells were cultured on glass slides in the wells of a 24-well plate (Rosmedbio, St. Petersburg, Russia) until they reached a subconfluent state. Each well was divided into three groups, with two slides per group: (1)The first group served as a negative control, with cells not incubated with the virus.(2)The second group served as a positive control, with cells incubated with the virus for the designated study time.(3)The third group was an experimental group, where cells were pretreated with K142 for 1 h and then incubated with the virus for the same duration as the control group.

The compound was initially dissolved in DMSO and adjusted to a final concentration of 100 µg/mL. For the positive and negative control wells, a culture medium containing a similar concentration of DMSO was used instead of the compound.

After specific time intervals, the cell culture was fixed: first, after a 2-h period of virus penetration into the cells, and second, after a 3-day period of syncytium formation. The cells were washed with PBS and then fixed in a 10% buffered formalin solution with a pH of 7.4 for 24 h. Subsequently, they were washed with PBS solution, and any non-specific binding sites of antibodies were blocked by incubation in a 2% BSA (bovine serum albumin) solution in PBS for 20 min.

Next, the cells were stained with mouse primary antibodies against RSV (dilution of 1:200) obtained as described in [13], washed three times with PBS, and incubated with a solution of rabbit secondary anti-mouse antibodies (dilution of 1:500) conjugated with a fluorescent label (Alexa 488). This step was followed by three washes with PBS and incubation with a Hoechst 33,342 solution (dilution of 1:500) to stain DNA (cell nuclei).

The coverslips with stained cells were then removed from the plate and mounted on slides using Fluoroshield™ containing triethylenediamine. The resulting preparations were analyzed using a Leica SP8 laser scanning confocal microscope equipped with a 60×/1.4 objective with oil immersion and a pinhole parameter of 1 AU. A 488 nm argon laser with a power parameter of 10% was used to excite Alexa 488 fluorescence. 

To record Alexa 488 fluorescence, a signal was captured in the wavelength range of 490–550 nm, utilizing a detector voltage of 750 V. Hoechst 33,342 fluorescence was excited using a 405 nm diode laser at a power setting of 50%. The resulting fluorescence signal was then recorded in the range of 410–500 nm, with a detector voltage of 800 V.

To quantify the number of virus particles bound to the cells, confocal images of the cell culture were analyzed using the ImageJ program. The images had a resolution of 1024 × 1024 pixels (246.03 × 246.03 µm). Five random fields of view were selected from each preparation to ensure an even distribution of cell nuclei, with the optical section passing through the middle of most of the nuclei.

The acquired images (n = 10 in each group) were stored in separate channels corresponding to the staining for nuclei (blue) and viruses (green). The green channel images were converted to 8-bit format, and a background subtraction algorithm was applied using a “Rolling ball radius” parameter of 50 pixels. Subsequently, the average pixel intensity was estimated to determine the fluorescence intensity of the secondary antibodies (Alexa 488) associated with the primary antibodies specifically binding to the viral particles on the cell surface. Statistical analysis of the results was performed using the GraphPad Prism 8 software.

### 4.3. Materials and Methods for In Vivo Experiments

#### 4.3.1. Reference drug (positive control)

Reference drug Ribavirin (positive control), Vertex, Russia. Ser #030222, reg. passport #896 from 08.04.2022. Dosage form: substance. Storage conditions: at a temperature of +4 °C. 

#### 4.3.2. Placebo (Negative Control)

Phosphate-buffered saline solution DPBS (Biolot, St. Petersburg, Russia). Dosage form: solution. Description: clear liquid. Packed in 450 mL plastic containers. Storage conditions: in the refrigerator at a temperature of +2 °C–+8 °C. 

#### 4.3.3. Experimental Animals

The study was conducted using 4–6-week-old Balb/c pure female mice. Each group consisted of 15 mice, with 10 animals used to determine viral load and changes in body weight in the lungs, and 5 animals used to study the histological picture of lung lesions. All experiments involving laboratory animals were approved by the Bioethics Commission of Smorodintsev Influenza Research Institute, St. Petersburg, Russia. 

The mice were obtained from the nursery Stolbovaya of the Scientific Center of Biomedical Technologies, Moscow Region. They were kept under standard conditions, following the methodological documents used in the research. 

#### 4.3.4. Housing of Laboratory Animals

The mice were housed in polycarbonate cages (type T3A, S = 1200 cm^2^) from BENEX a.s., Czech Republic. They were kept in groups of 15 individuals on bedding made of wood pellets from Laboratorkorm LLC. The cages were covered with steel lattice covers with a stern recess. 

The mice were provided with ad libitum access to granulated food from (Laboratorkorm, Moscow, Russia). Water purified by reverse osmosis using a Millipore RiOs 30 water treatment plant was also given ad libitum in standard drinking bowls with steel spouts. Wood pellets from (Laboratorkorm) were used as bedding material. 

The mice were kept in separate rooms in the vivarium of the Smorodintsev Influenza Research Institute. The ambient conditions were controlled (18–24 °C and relative humidity of 50–80%), and the photoperiod was 12 h night–12 h day under artificial lighting with fluorescent lamps. The care and maintenance of the animals were carried out according to SOPs, adopted by the FSBI Smorodintsev Influenza Research Institute 

Animals were randomly assigned to groups based on body weight, ensuring that each individual’s weight fell within ±10% of the mean value. Each animal in the group was assigned a unique number.

Euthanasia: After the completion of drug administration and non-invasive tests, planned euthanasia was conducted using CO_2_.

#### 4.3.5. Administration Procedure 

The studied compounds and controls were administered to animals in accordance with the treatment-and-prophylactic scheme. This involved administering the samples one day before infection, on the day of infection, and 1, 2, 3, 4, and 5 days after infection. The compound K142 was given orally using a gastric tube three times a day at a dosage of 50 mg/kg for Group 1. For Group 2, K142 was administered intranasally under light ether anesthesia in a volume of 15 μL at doses of 10 mg/kg, and, for Group 3—1 mg/kg. 

To control the specificity of the pathological process, a reference drug, ribavirin, was used in Group 4. This was administered orally using a gastric tube in a volume of 0.2 mL at a dosage of 50 mg/kg. In Group 5, a negative control (Placebo), saline was administered orally in a volume of 0.2 mL. 

For a clearer understanding of the experiment, please refer to Table 2 which presents the general scheme.

#### 4.3.6. Virus Introduction Procedure

Mice were intranasally infected with the RSV-A virus under light ether anesthesia. A volume of 50 μL, containing a viral titer of 5.5 lgTID_50_, was administrated. Following infection, the animals were observed for a period of 5 days.

#### 4.3.7. Body Weight Control in Mice

To monitor the weight changes, mice were weighed daily, starting from day-1 (the day before infection). The weighing process was conducted during the midday, before the second drug injection.

#### 4.3.8. Organ Harvesting and Preparation of Homogenates

On the 6th day after infection, all animals in each group were euthanized. The lungs were carefully extracted from the mice. In each group, 5 mouse lungs were fixed in formalin for future histological studies. The remaining 10 mouse lungs were placed in sterile, pre-weighed eppendorfs to determine their weight. The lungs were then homogenized in phosphate-buffered saline DPBS (Biolot, St. Petersburg, Russia) using a Tissue Lyser II device (Qiagen, Germantown, MD, USA) within the same eppendorfs. The resulting organ suspension was utilized for assessing the viral load through the application of sandwich ELISA.

#### 4.3.9. Determination of Viral Load in the Lungs by Sandwich ELISA

The viral load present in the lungs was determined using a sandwich ELISA method. To carry out the enzyme immunoassay, Microlon High Binding 96-well plates (#655061, GreinerBio-One, Kremsmünster, Austria) and PST-60HL-4 thermoshaker plates (BioSan) were utilized.

In the first step, the capture monoclonal antibody 7B12 specific to RSV F protein [13] was diluted in 1× phosphate monophosphate buffer solution (FMSB) and added to each well (100 µL) of the plate. The plate was then incubated overnight at +4 °C.

Afterward, any unbound antibodies were removed by washing the wells with a PBS-T solution (0.05% Tween-20). To prevent non-specific binding, the wells were blocked with a solution of 5% milk (Blotting-Grade Blocker, #1706404, Bio-Rad, Hercules, CA, USA) dissolved in PBS-T. The blocking step was carried out at 37 °C for 1 h.

Next, the analyzed samples, standards, or a negative control solution (PMSB-T) were added to the plate and incubated at 37 °C for 2 h. A standard suspension of RSV with a concentration range from 5000 to 80 ng/mL was used. The concentration of the viral suspension was determined using the Lowry method.

Following the sample incubation at 37 °C for 2 h, the plate was washed three times with the PMSB-T solution for 2 min each time. Subsequently, a conjugate of biotin isothiocyanate (BITC) tagged monoclonal antibodies (4F2) specific to RSV F protein was added (100 μL per well) and incubated at 37 °C for 1 h.

The presence of the biotinylated antibodies bound to the antigens was detected using a streptavidin-horseradish peroxidase conjugate (streptavidin-HRP) (R&D Systems, Wuhan, China), which was diluted 1:1000 in PMS-T. The plate was incubated at 37 °C for 30 min. 

The peroxidase reaction was demonstrated by adding 100 µL of substrate mix containing 9 parts of Solution A and 1 part of Solution B from the TMB Peroxidase EIA Substrate Kit (#1721067, Bio-Rad, Hercules, CA, USA) to each well of the plate. After stopping the reaction by adding 100 µL of 1N sulfuric acid to each well of the plate, the absorbance was measured at 450 nm (OD_450_) and 655 nm (OD_655_) using a CLARIOstar multimodal microplate reader (BMG Labtech, Ortenberg, Germany).

A standard sample of the virus was cultivated on HEp-2 cell culture. Following virus accumulation, purification was accomplished through differential centrifugation in a sucrose gradient. The concentration of the virus (measured by total protein) was determined using the Lowry method.

#### 4.3.10. Histological Examination of Lung Lesions

To prepare lung fragments for histological studies, they were fixed in a 10% buffered formalin solution with a pH of 7.4 for 48 h. After obtaining the required sections, routine histological processing was performed using a Histo-Tek VP1 histoprocessor (Sakura, Tokyo, Japan), followed by encapsulating the samples in paraffin blocks. The resulting 2 µm thick sections were stained using a hematoxylin and eosin solution. Microscopy analysis was conducted using a LEICA DM1000 light microscope. Measurements and image capture were carried out using the ADF Imager.

#### 4.3.11. Evaluation of the Specific Pharmacological Activity of Samples In Vivo

The main criterion for evaluating antiviral activity is the reduction in viral load in the lung tissue of mice in the group treated with the drug in comparison with the placebo group. Secondary criteria for assessing antiviral activity: the influence of the studied drug on weight dynamics of infected animals—a decrease in weight indicators of the lungs of infected animals—reduction in signs of the infectious process in the lung tissue during histological examination 

#### 4.3.12. Analysis of Results

Evaluation of the specific pharmacological activity of samples in vivo involves assessing the antiviral activity by measuring the reduction in viral load in the lung tissue of mice treated with the drug compared to a placebo group. Additional criteria for evaluating antiviral activity include the impact of the drug on the weight dynamics of infected animals and a reduction in signs of the infectious process during histological examination of the lung tissue.

The analysis of the results was conducted using Microsoft Excel and GraphPad Prism 8.0 software. To visually represent the data on the relative drop in body weight, the percentage value of body weight relative to the weight on the day of infection (day 0) was calculated for each animal. The mean value for the group was determined, and a curve showing the dependence of the group mean on the days after infection was plotted.

For the analysis of Sandwich ELISA results, MARS Data Analysis, Microsoft Office Excel, and GraphPadPrism 8 programs were used. To account for the background signal, the difference between the corresponding optical density values (OD_450_ and OD_655_) was calculated for each well. The mean value of all negative controls, plus three standard deviations, was established as the threshold value.

To determine the significance of differences between group means, the one-way ANOVA analysis of variance was used for group comparison. Following this, the Dunnett test was applied for post hoc pairwise comparisons with the Placebo group. 

### 4.4. Calculation Methods

The geometric parameters of the ligand–protein complex, obtained through molecular docking as described in [5], were used to construct a model for molecular dynamics simulation. Trimer-ligand complexes were selected for two lead compounds, and subsequent molecular dynamics simulations were performed.

Using the structure of the full-length F protein, the transmembrane domain (amino acid residues 525–550) was embedded within a phosphatidylcholine (POPC) membrane. Phosphatidylcholine is a common constituent of viral cell membranes [14,15]. The complexes were positioned within an orthorhombic simulation box measuring 15 × 35 × 55 Å, filled with a 0.15 M aqueous NaCl solution. The TIP3P solvent model was employed, and counterions were added to maintain system neutrality. The simulation was conducted in the NPγT thermodynamic ensemble at a temperature of 310 K (37 °C) for a duration of 50 nanoseconds.

The system preparation protocol involved initial minimization and equilibration of the system components. These calculations were performed using the Desmond program [16], which is integrated into the Small Molecules Drug Discovery Release 2021-4 software package.

## Figures and Tables

**Figure 1 ijms-24-12933-f001:**
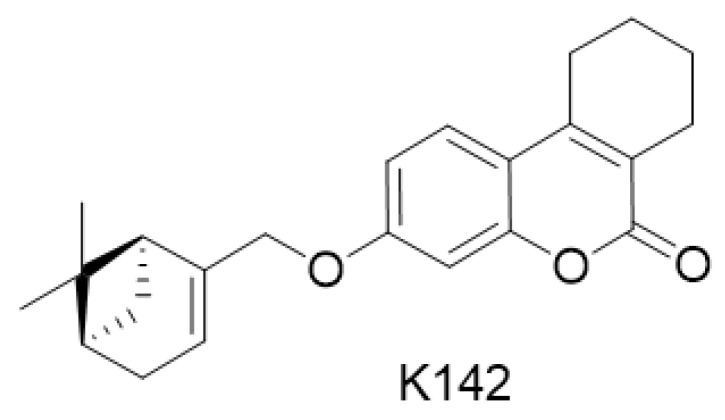
Structure of compound under study K142 (((3-(((1S,5R)-6,6-dimethylbicyclo [3.1.1]hept-2-en-2-yl)methoxy)-7,8,9,10-tetrahydro-6H-benzo[c]chromen-6-one))).

**Figure 2 ijms-24-12933-f002:**
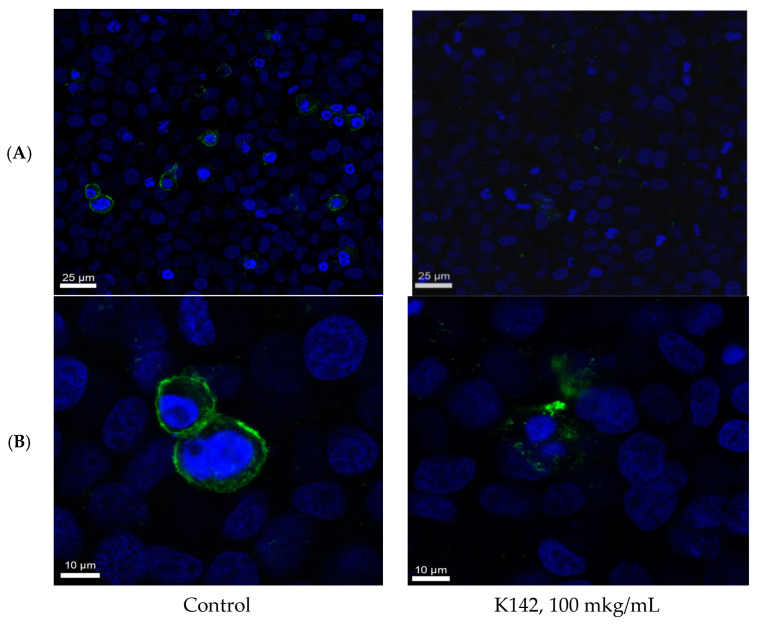
Fluorescent confocal microscopy observations of Hep2 cells at 2 h after incubation with RSV. The RSV was stained with primary Abs against RSV and secondary Abs conjugated with Alexa 488 (green). Cell nuclei were stained with Hoechst 33,342 (blue). (**A**), scale bar = 25 μm; (**B**), scale bar = 10 μm.

**Figure 3 ijms-24-12933-f003:**
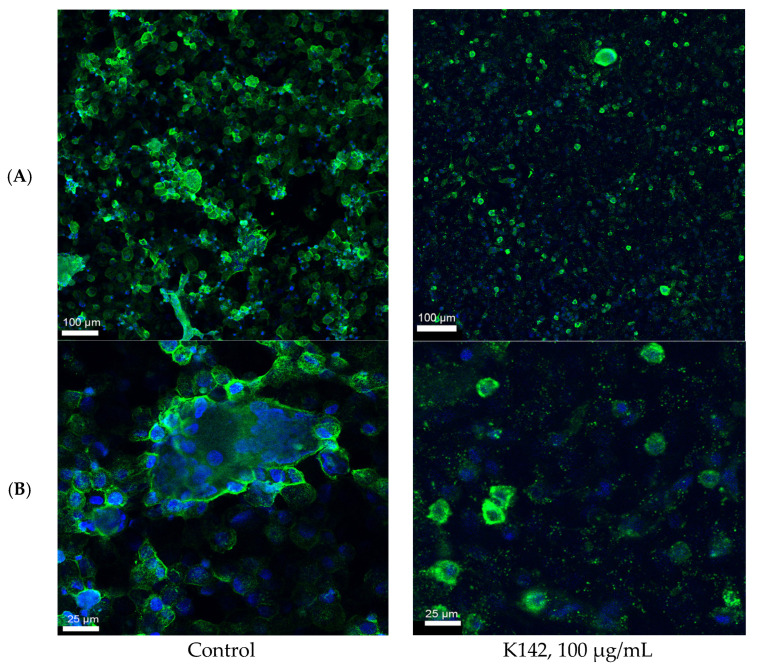
Fluorescent confocal microscopy observations of Hep2 cells at day three after incubation with RSV. The RSV was stained with primary Abs against RSV and secondary Abs conjugated with Alexa 488 (green). Cell nuclei were stained with Hoechst 33,342 (blue). (**A**), scale bar = 100 μm; (**B**), scale bar = 25 μm.

**Figure 4 ijms-24-12933-f004:**
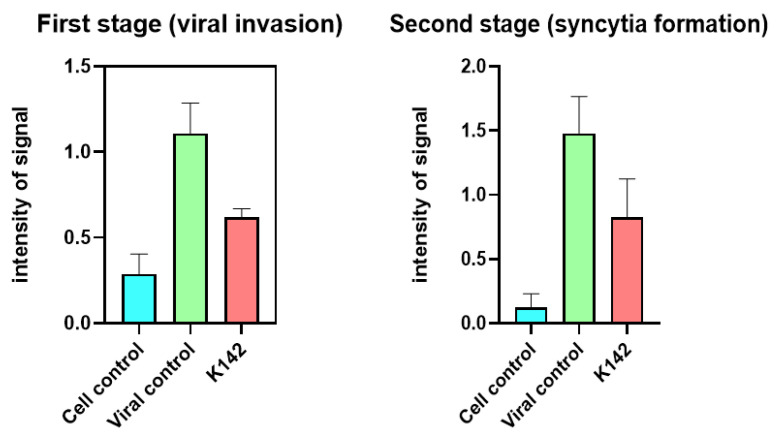
Results of the statistical analysis of compound K142 effects on the intensity of viral antigen signal observed by confocal microscopy at different stages of the RSV life cycle.

**Figure 5 ijms-24-12933-f005:**
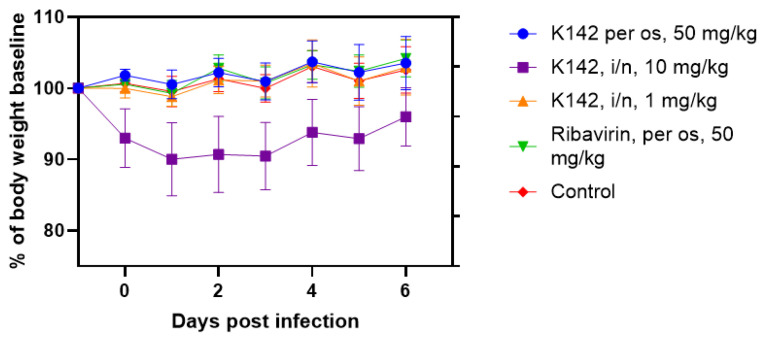
Dynamics of changes in body weight of Balb/c mice infected with RSV under the conditions of application of the studied drugs.

**Figure 6 ijms-24-12933-f006:**
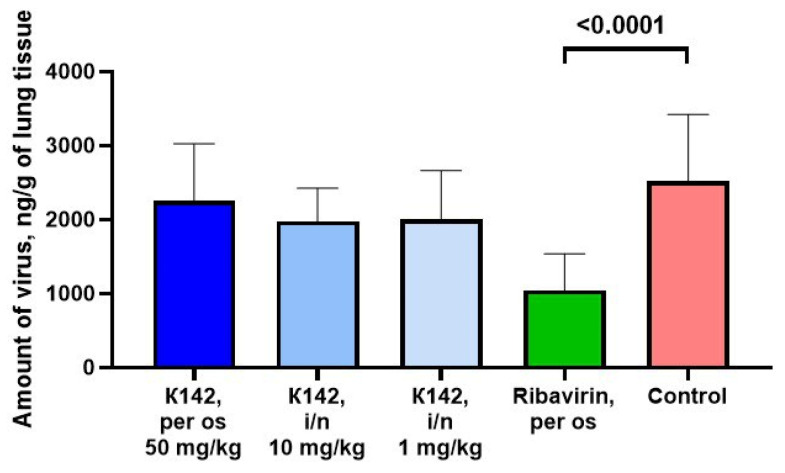
The impact of compound K142 on the viral load in the lung tissue of Balb/c mice infected with RSV.

**Figure 7 ijms-24-12933-f007:**
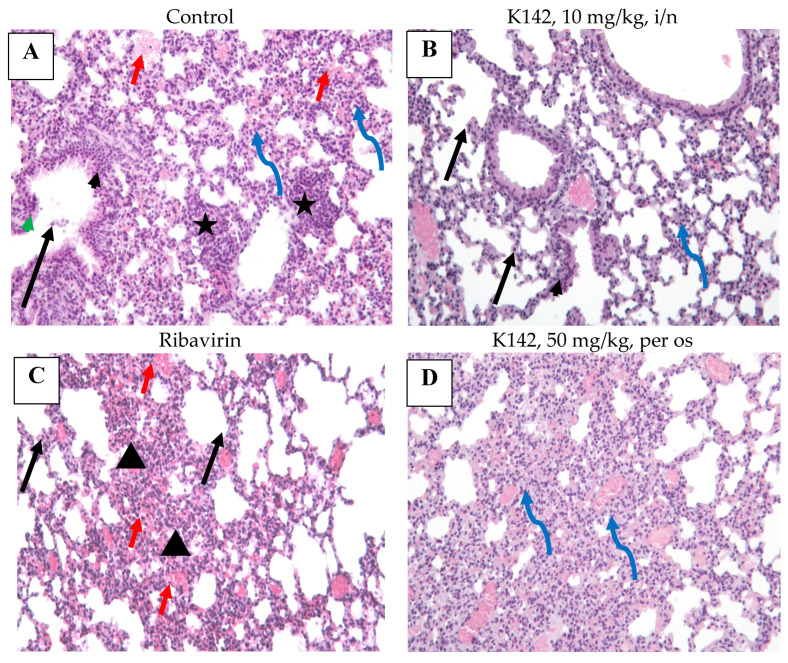
The effect of compound K142 on the histological features of bronchiolitis in Balb/c mice lung tissue infected with RSV. Hematoxylin-eosin, 200×. (**A**): acute lung injury: tissue appears compressed, respiratory space reduced due to the massive lymphohistiocytic alveolar wall infiltration, swelling and hemorrhage (blue curved arrow), diffuse type II alveolocyte hyperplasia is presented, foci of lymphoid infiltration (black star) turning up in the paravasal septa; airways: small bronchi respiratory epithelium appears hyperplastic (black arrowhead), foci of epithelium atypia consisting of cells with abundant foamy cytoplasm and highly-basophile nuclei, crowding in the basal region (syncytial transformation, green arrowhead), local desquamation of atrophic epithelium (black arrow), some focal intraluminal hemorrhage noted (red arrow). (**B**): minimal pathology lung tissue: alveolar septa exhibit scattered type II alveolocyte hyperplasia (black arrow) and minor lymphohistiocytic infiltration (blue curved arrow). Minor respiratory epithelium hyperplasia noted in the bronchiole (black arrowhead). Marked microcirculatory hyperemia. (**C**): atelectasis: focus of airless lung tissue composed of swollen, thickened, infiltrated alveolar septa (black triangle). Marked lymphohistiocytic infiltration of septa in combination with wide-spread intraalveolar hemorrhages. Scattered type II hyperplasia presented across the section (black arrow). Note pronounced microcirculatory hyperemia with erythrocyte adhesion, forming “columns”. (**D**): pneumonitis: respiratory space markedly thinned due to thickened, hyperplastic alveolar walls (blue curved arrow) and intraluminal hemorrhage. Infiltrating cells are predominantly macrophages and lymphocytes with admixed polymorphonuclear leukocytes. Interalveolar septa forming dense atelectasis across whole lung section. Microcirculatory vessels appear hyperemic, fulfilled with erythrocyte casts. Scattered type II hyperplasia present outside the infiltrate area.

**Figure 8 ijms-24-12933-f008:**
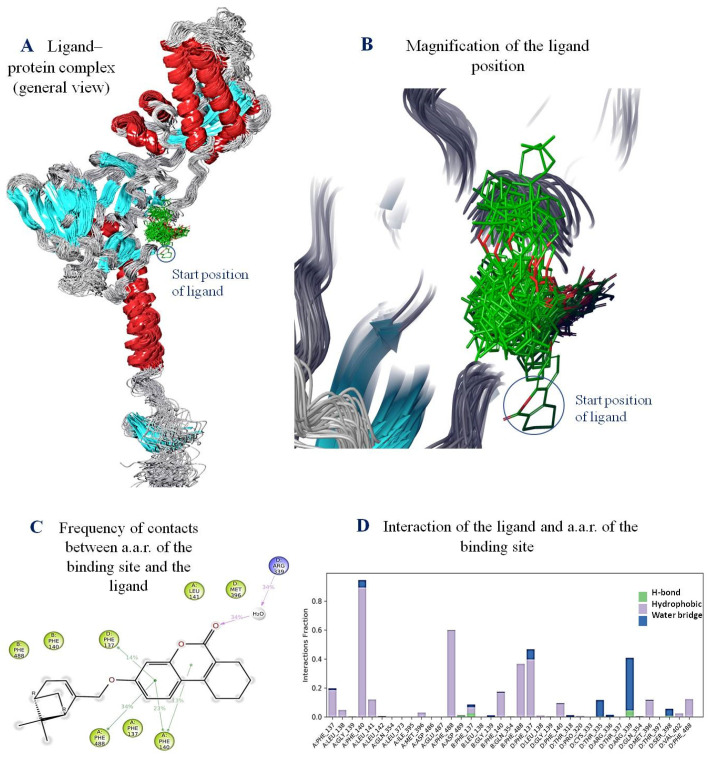
Results of molecular dynamics simulations: (**A**)—superimposition of geometric parameters of the ligand–protein complex (to facilitate visualization, only the F-protein protomer is shown) corresponding to 50 frames in increments of 100, ligand molecules are shown in green; (**B**)—magnification of the ligand position image depending on the simulation time; (**C**)—frequency of contacts between a.a.r. of the binding site and the ligand (in %): π-π stacking interactions are shown by green lines, hydrogen bonds are shown by a purple arrow; (**D**)—a histogram of interaction of the ligand and a.a.r. of the binding sites, the time of contact (in fractions) of the ligand with a.a.r. is plotted along the Y axis.

**Table 1 ijms-24-12933-t001:** Average values of the level of viral antigen in the lung tissue of Balb/c mice infected with RSV under conditions of studied drug application. *p*—the result of applying the Dunnett criterion when comparing data with the control group.

Compound	Method of Administration and Dose	The Amount of Antigen in the Sample, ng/mL	The Amount of Antigen in Terms of 1 g of Lung Tissue	*p*
K142	Per os, 50 mg/kg	337 ± 134	2257 ± 771	0.7993
	intranasally, 1 mg/kg	300 ± 48	1978 ± 445	0.2330
	intranasally, 10 mg/kg	294 ± 86	2000 ± 661	0.2657
Ribavirin	Per os, 50 mg/kg	169 ± 88	1044 ± 493	<0.0001
Control	Per os	407 ± 162	2520 ± 900	-

**Table 2 ijms-24-12933-t002:** Design of the experiment to assess the protective activity of the drug K-142 in Balb/c mice respiratory syncytial infection model.

Group	Compound	Administration	Dose	Scheme of Drug Administration
1	K-142	Per os	50 mg/kg	3 times a day
2	K-142	i/n	1 mg/kg	3 times a day
3	K-142	i/n	10 mg/kg	3 times a day
4	Ribavirin	Per os	50 mg/kg	1 time a day
5	Placebo	Per os	—	3 times a day

## Data Availability

Research data can be found in the Russian Science Foundation report for grant 21-13-00026 (https://grant.rscf.ru/site/user/forms?rid=000000000000003093287-3), accessed on 14 July 2023, information is available for registered users only.

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
