# Peer review of "Identification and Study of the Action Mechanism of Small Compound That Inhibits Replication of Respiratory Syncytial Virus"

_ijms, 2023, doi:10.3390/ijms241612933_

Round 1

Reviewer 1 Report

In this manuscript, Shtro et.al describe the outcomes of in vivo testing of compound K142 which is a potential anti-RSV drug. RSV remains a significant clinical challenge for both pediatric and geriatric patients worldwide and thus novel treatments and vaccination strategies are needed to manage RSV disease.

The authors show that K142 does reduce RSV infectivity and syncytia formation in vitro, interacts with RSV F protein and does not induce resistance. However, in vivo activity of K142 was poor and the authors recommend further modifications to the K142 backbone or improvements to its bioavailability.

The manuscript is poorly written; experimental details (reagent concentrations, volumes, time of addition, infection MOI, number of repeats) are either missing or difficult to find in the manuscript.

Following are the major comments on the manuscript.

1.    In the results section, please start by describing what was done before you state the observations. For example, in Figure 2. Was the drug added before infection, along with the virus or after infection ? what was the concentration of the drug used, was it dissolved in water or another organic solvent like DMSO? In case it was, did the authors test DMSO as a mock control? All these details should be clearly mentioned in the material methods and briefly stated in the figure legend.

2.    Line 466- what was the primary antibody used, was it against whole virus or F / G? what titer was used for staining for both primary and secondary antibodies ?

3.    Figure 6. Graphs have no axis labels. How was full-genome coverage of collected RSV strains determined? Experimental details of sequencing are missing completely. Pl. include those in the manuscript.

Experimental details in materials and methods need substantial improvement. All the animal housing, feed and related sections can be combined into one section. Results should be written in the format; what is the question we are trying to answer, how did we do that  and what did we get.

Author Response

Thank you for reviewing our manuscript. Please find attached the file containing our responses.

Reviewer 2 Report

In this manuscript, the authors describe the mechanism and in vivo activity of a chemical compound against RSV.

There are several flaws in this article, which need to be revised.

Major comments:

·        It is not clear how efficacious K142 is and why you have chosen to continue with this molecule. I am missing  an IC50 value of K142, can the compound completely neutralize the virus (and at what concentration)? Please add this information. Line 54-55 insinuates that you have more potent structural analogues? Yet it is not clear in this publication (ref 6) which compound is K142.

·        To prove that the compound has an effect on RSV F, you can test if it inhibits syncytium formation of RSV F-transfected cells.

·        K142-resistant strain evaluation. It is completely not clear how you have performed this experiment, as there is no information in the materials and methods.  Without prior knowledge of the neutralization curve (IC50) it is not possible to perform this assay, as you should know at what concentration only a very limited amount of virus can still grow. You mention that the upper concentration was 3 µg/ml as there were no syncytia at 4 µg/ml. But you do not need syncytia for viral proliferation. In the assay of section 2.1 you can see that the virus still grows at 100 µg/ml! so at 3 µg/ml you do not apply enough pressure on the virus to mutate.  It is also not clear how the results of figure 5 and table 1 have been established. Is this a neutralization assay? Have you started from the same amount of virus? It seems like you did not as there is more virus in the 'control strain' compared to the 'resistant strain' at 0 µg/ml. So you cannot conclude here that there is a difference in resistance. There also cannot be a difference because the viruses are exactly the same, like you have proven with the sequencing.  If the compound does not lead to a complete neutralization, it is very hard to do a resistant strain evaluation as wild type virus will always remain. If it does lead to a complete neutralization, you should add as much compound as possible before you obtain this complete neutralization, and then transfer the supernatant of wells in which only a very limited amount of virus managed to grow (test by performing plaque staining).  So for this article, either repeat this experiment or remove this section as this does not bring any information for the reader. You cannot conclude that the compound does not induce resistance as the experiment was not performed well.  Also, what is Ig TID50?  IC50 determinations based on PFU is much more clear.

·        In vivo experiment. It is not clear why you consider doing an in vivo experiment testing a compound with such weak activity.  15 µl IN administration seems like a very low volume, possibly this did not reach the lung but remained in the nose, how is this administrated? If the viscosity was an issue, why did you not dilute this sample and gave a higher volume? What titer of virus did the mice receive? This is essential information to understand the results. The amount of antigen in the lung samples is not easy to interpret, please provide PFU/lung.  

·        Molecular dynamics cannot be used to confirm a target. This can only be used as an indication as where it could bind. To confirm, a crystallization study should be performed.

·        The  discussion if very limited, and does not include a comparison to other chemical compounds. What makes this compound interesting compared to competitors? Please elaborate.

·        line 273: the compound under study does not seem promising to me.

Minor comments:

·        Two vaccines have been accepted. So it is not true anymore that no vaccines are available. Please adapt your abstract and introduction.

·        Please introduce your paragraphs more clearly. Line 64 starts immediately with describing your experiment without telling why  you are doing this experiment.

·        Scale bars of fig 2 are are not readable, scale bars of fig 3 are missing.

·        line 64 ' on the first or second day' be more precise.

·        Paragraph 2.1. The set-up of the experiment is not clear, was the compound added before infection or during infection? Was the inoculum/ compound washed away? Please clarify in the text

·        Line 72, 'RSV-tagged alexa fluor 488'? not correct, the dye was off course not tagged with virus. this should be 'observations of Hep2 cells stained with ...' please also adapt in figure 3.

·        Make your captions (from all figures!) more informative. You should be able to understand a figure without reading the main text.

·        Line 88, there is no table. 'in the control of the virus', you probably mean just 'the control'.

·        Please include a diagram of the mouse experiment (times of administration and infection).

·        How are the mice sedated for the administrations

·        Please make figure 7 in graphpad, this excel graph does not look professional.  Also provide standard deviations.

·        Not clear what the p-values mean in table 2, please explain in the caption

·        Figure 10 is not clear and well readable, please adapt.

·        figure 9. I am not trained in looking at lung sections, please add more information to these pictures, point to inflammatory foci? This would be informative for most readers.

·        Overall language should be improved, if possible edit the article by a native speaker.

·        Some examples

o   line 18: It's should be its

o   line 31-32, major cause of ill-being and unwellness (?) as well as disease in humans (so newborns and young children are not humans?)

o   ALRI is the most important cause of mortality in children

o   Line 34: RSV causing SARS? you probably mean ALRI

o   Line 175: Lungs were opened?

o   The term 'chemotherapy' is generally only used in the field of cancer. Please adapt

Author Response

Thank you for reviewing our manuscript. Please find attached the file containing our responses

Round 2

Reviewer 2 Report

The manuscript has been much improved. 

If possible, please also explain the concept of lgTID50 (google has not been helpful). The first steps in the cell elisa are not explained (which plate was used, how long after inoculation you perform the elisa). 

 There are also some still some typos such as 'ELIZA' in stead of elisa. Always refer to 'RSV' in stead of writing the full name, after the first explanation of the abbreviation. 

Author Response

Thank you for the reviewing our manusript. The text was revised.

Also, unswering to your question about TID50. It is 50% tissue infectious dose, is a measure of the amount of virus required to infect 50% of cell culture wells. lgTID50,refers to the logarithm (base 10) of TID50. It is calculated using the Reed and Muench method:

log10 50% end point dilution = log10 of dilution showing an infectivity next above 50% - (difference of logarithms × logarithm of dilution factor).

For a more detailed understanding of the calculation method, you can refer to the article available at: [https://www.ncbi.nlm.nih.gov/pmc/articles/PMC7973348/].

Some authors may use the term TCID50, which stands for 50% tissue culture infectious dose, as an alternative term to describe viral titer. These two terms are synonyms.